# Advances in AI for Protein Structure Prediction: Implications for Cancer Drug Discovery and Development

**DOI:** 10.3390/biom14030339

**Published:** 2024-03-12

**Authors:** Xinru Qiu, Han Li, Greg Ver Steeg, Adam Godzik

**Affiliations:** 1Division of Biomedical Sciences, School of Medicine, University of California Riverside, Riverside, CA 92521, USA; xinru.qiu@ucr.edu; 2Department of Computer Science and Engineering, University of California Riverside, Riverside, CA 92521, USA; han.li001@email.ucr.edu (H.L.); greg.versteeg@ucr.edu (G.V.S.)

**Keywords:** AlphaFold2, cancer, drug discovery, artificial intelligence, generative AI

## Abstract

Recent advancements in AI-driven technologies, particularly in protein structure prediction, are significantly reshaping the landscape of drug discovery and development. This review focuses on the question of how these technological breakthroughs, exemplified by AlphaFold2, are revolutionizing our understanding of protein structure and function changes underlying cancer and improve our approaches to counter them. By enhancing the precision and speed at which drug targets are identified and drug candidates can be designed and optimized, these technologies are streamlining the entire drug development process. We explore the use of AlphaFold2 in cancer drug development, scrutinizing its efficacy, limitations, and potential challenges. We also compare AlphaFold2 with other algorithms like ESMFold, explaining the diverse methodologies employed in this field and the practical effects of these differences for the application of specific algorithms. Additionally, we discuss the broader applications of these technologies, including the prediction of protein complex structures and the generative AI-driven design of novel proteins.

## 1. Introduction

ChatGPT, DALL-E, and other tools, driven by recent revolutionary advances in Artificial Intelligence (AI) technology, have captured widespread attention due to the expanding capabilities of AI, with the promises and potential threats they bring to our society. However, AI-driven breakthroughs in various scientific fields have been in progress for some time. This review delves into a remarkable transformation within structural biology, catalyzed by the introduction of the AlphaFold2 (AF2) deep neural network algorithm in 2021 and followed by other algorithms. Together, these tools have effectively resolved a long-standing challenge in structural biology: the generation of atomic-level models for protein structures from sequence information alone [1,2,3]. This review seeks to investigate the extent to which these breakthroughs in protein structure prediction have influenced the drug discovery process, with an initial focus on cancer research, and also discuss how choices in the architecture and assumptions made by specific algorithms affect and differentiate their results.

The process of drug discovery is frequently marked by inefficiency, underscored by rising expenses, prolonged timeframes, and a high frequency of failures. Only a small fraction of drug candidates make it to clinical trials, and many fail as late as in Phase 3, resulting in an overall success rate of about 10–20% in clinical drug development [4]. Estimations of the overall expenses for research and development prior to product launch range from $161 million to $4.54 billion in 2019 U.S. dollars per successful drug [5] (Figure 1). This ineffectiveness is, in part, due to our incomplete understanding of human biology, especially in the context of disease processes; a dearth of actionable targets for treatment; and our limited understanding of the varied responses to disease in diverse populations [6]. Further complicating this process is the inadequacy of preclinical models that accurately represent the disease and the constraints of overly simplistic disease models, which together amplify the difficulties in grasping the complexity of human systems. Lack of high quality structural models of drug targets, a main problem addressed by AlphaFold, is only one of the challenges in drug discovery. However, as we show in this review, AI is also making rapid progress in addressing other bottlenecks in drug development.

Traditionally, the three-dimensional structures of proteins are deciphered using labor-intensive and costly experimental methods like X-ray crystallography, nuclear magnetic resonance (NMR), and cryogenic electron microscopy (cryo-EM). While invaluable, these techniques are limited by speed, cost, and applicability to only certain protein structures. In contrast, recent advancements in protein structure prediction, culminating in AF2, have dramatically expanded our capabilities, complementing and occasionally surpassing experimental approaches.

The AF2 breakthrough has been quickly followed by other AI tools such as RoseTTAfold [7], ESMFold [8], and OpenFold [9]. ProGen [10], ProteinMPNN [11], EvoDiff [12], and RFdiffusion [13] extend the AI capabilities to novel protein design, as does DiffDock [14] to molecular docking. These and many other rapidly developing tools apply novel algorithms and AI architectures, each with unique strengths and weaknesses. Here we focus not so much on the comparison of their predictions, but on the differences in their algorithms and approaches and the resulting optimal applications.

## 2. Protein Structure Prediction In Silico before AlphaFold

In the period preceding the advent of AlphaFold, the process of protein structure prediction generally encompassed several distinct stages, as outlined in the following discussion (Figure 2).

### 2.1. Homology and Comparative Modeling

Homology modeling predicts a protein’s 3D structure using the structure of a homologous protein. It involves four steps: identifying a homologous protein with a known structure (target identification), aligning the target with the template sequence (alignment), constructing a model of the target protein from aligned regions (model building), and enhancing the model’s accuracy and stability (model refinement). Improvements in distant homology recognition and alignment between distant homologies are exemplified by the HHpred algorithm and the accompanying suite of programs [15,16]. Predictions of protein contact maps from coevolution patterns approached this problem from another angle [17], enhanced by the first applications of deep learning neural networks [18]. In the late 2010s tools such as Rosetta [19] and I-Tassser [20] crossed the line from homology to comparative modeling [21]. Rosetta achieved this by using smaller elements of known structures and a combination of energy-like scoring function and empirical folding rules. I-TASSER (Iterative Threading ASSEmbly Refinement)’s similarity uses a combination of template-based modeling and fragment assembly. Other tools’ similarity has started approaching the level of ab-initio protein structure prediction [21]. These advances directly led to the development of AlphaFold and the following AI approaches to protein structure prediction.

### 2.2. Structure Validation

Structure validation ensures that the predictions are accurate and plausible. Tools like [22] analyze the geometry of structural features and verify the dihedral angles in the Ramachandran plot. Energy based evaluations, such as ANOLEA [23], assess potential energy to evaluate the correctness of folding. Finally, predicted structures can be compared to the experimental data, if such are available. Such comparisons can be used to benchmark the prediction methods and establish expected accuracy, but cannot be used to evaluate predictions for proteins with no known experimental structures. However, functional predictions based on the predicted 3D structures, such as identity of active site or interaction interface residues, can be tested in vitro, thus indirectly confirming the structure prediction.

## 3. Existing Protein Structure Data Sets and Their Applications

Existing protein structure data sets play a pivotal role in protein bioinformatics (Table 1). Protein structures elucidated through experimental methods by various structural biology research groups are submitted to the Research Collaboratory for Structural Bioinformatics (RCSB) Protein Data Bank (PDB) [24]. The practical applications of AI-based structure predictions have been made much easier by the development of the AlphaFold Protein Structure Database, which offers precalculated predictions for over 200 million protein structures. Integration of the AlphaFold2 and the UniProt databases extended access to protein structural information to a broad community of biologists [25,26]. The ESM Metagenomic Atlas contains predictions for over 700 million protein structures from various microorganisms found in environments such as soil, seawater, and the human gut. This comprehensive collection of predicted structures provides valuable insights into the metagenomic landscape [8]. These data sets collectively support a broad range of research studies and applications, including developing and evaluating machine learning models, advancing our understanding of protein biology, and facilitating drug discovery efforts.

## 4. Disease Understanding—Examples of the Applications of AI-Based Structure Predictions

### Understanding Pathogenic Mutations

AF2 protein structure predictions can help identify pathogenic missense variations in hereditary cancer genes. In a study by Karakoyun et al. [30], AF2-predicted structures and five protein stability predictors were used to evaluate the pathogenicity of more than a thousand missense variants from ClinVar and a breast cancer patient cohort. Their findings indicated that protein stability predictors show moderate effectiveness in identifying pathogenic variants. However, the AF2 confidence score, pLDDT, demonstrated a superior ability to predict pathogenicity, highlighting AF2’s potential in pinpointing genetic variations linked to cancer.

AF2 can also help us understand the role of the paralogs of disease proteins. For example, dysfunction of human diacylglycerol kinase (DGK) is linked to multiple diseases, including cancer and autoimmune disorders. However, the exact mechanism of how DGK dysfunction contributes to the development of these diseases is not fully understood due to the lack of high-resolution structures for any of the 10 human DGK paralogs. In a recent study [31], the researchers used AF2 to predict the three-dimensional structures of all the human DGK paralogs and conducted structural alignment of the predictions to reveal the conserved domains and their spatial arrangement relative to each other. The study also used docking studies to corroborate the existence of a conserved ATP-binding site between the catalytic and accessory domains and to investigate the spatial arrangement of DGK with respect to the membrane.

AF2 can aid drug discovery by accurately predicting protein 3D structures and identifying potential allosteric binding sites. Allosteric drugs, which bind the allosteric rather than the active sites, can induce conformational changes in proteins, affecting their activities. This enables the design of more effective drugs that can synergize with traditional orthosteric drugs to enhance efficacy. A study from Nussinov, R., et al. [32] illustrated how allosteric drugs can alter the conformation of an active site that a drug-resistant mutation has created, permitting a blocked orthosteric drug to bind. This suggests that a combination of allosteric and orthosteric drugs can be more effective than either drug type alone. In another study from Weng, Y., et al. [33], AF2 was used to predict the protein structure of WSB1. The predicted structure was then optimized using molecular dynamics simulations and validated using software. After that, virtual screening was performed using AutoDock-GPU and Glide to filter compounds using ligand- or structure-based methods. Finally, four compounds with different compound scaffolds were selected as potential inhibitors of WSB1.

In a recent development, AlphaMissense, a computational tool devised by Google DeepMind, was shown to correctly assess the pathogenic potential of missense variants [34]. By utilizing the structural insights from AlphaFold, AlphaMissense evaluates the effects of mutations on the functionality of proteins. In the realm of cancer drug discovery, this tool holds significant promise in aiding researchers to efficiently select genetic mutations for in-depth study. This could expedite the process of identifying novel drug targets. Furthermore, AlphaMissense has the potential to enhance our comprehension of less-explored segments of the genetic code, especially genes that play crucial roles in human health but whose functions are yet to be fully understood.

## 5. Target Identification

The next step after understanding the molecular mechanism of disease is identifying targets for therapeutic intervention. Again, knowledge of the structure of proteins involved in pathways or networks mutated or modified in cancer is an important step in identifying best drug targets. Understanding the molecular mechanisms of disease at the molecular level, including the functional, interactive, and mechanistic implications of gene product alterations, is essential for developing targeted therapeutic strategies for cancer. By modeling these aspects, researchers can evaluate and compare different strategies to correct the adverse outcomes caused by gene mutations. Such molecular models are instrumental in the design of effective cancer therapies [35].

### 5.1. Prediction of Structures of Protein Complexes

Accurate prediction of protein complex structures is vital for cancer drug discovery, offering insights into the molecular mechanism of signal transduction (where physical interactions between up- and down-stream elements of the signaling pathway are used to pass on the signal) or indirect mutation effects (when a mutation in another element of the complex is modifying the function of a critical protein). Structure-based approaches are instrumental in developing specific and effective drugs, as well as in addressing drug resistance issues. They also support personalized treatments by identifying unique vulnerabilities in cancer cells of specific patients and aid in minimizing drug side effects and interactions.

In a study by Zhang, J., et al. [36], AF2 was used to predict the structures of protein complexes involved in cancer protein–protein interactions (PPIs). The researchers utilized AF2 to explore the protein–protein interactome associated with cancer, identifying 1798 potential protein–protein interactions (PPIs) related to cancer driver proteins. These proteins play roles in various cellular functions, including transcription regulation, signal transduction, DNA repair, and cell cycle processes. For the predicted binary protein complexes, they constructed spatial models, revealing that 1087 of these complexes had not been previously characterized in terms of their 3D structures. In addition, the top AF2 contact probability between residues of a protein pair can be used to distinguish true PPIs from false ones in yeast.

Vasoactive intestinal peptide receptor 2 (VIPR2), a class B G-protein-coupled receptor, plays a role in numerous physiological processes through its interaction with vasoactive intestinal peptide (VIP) and pituitary adenylate cyclase-activating polypeptide (PACAP). VIPR2 has garnered interest as a potential therapeutic target in the fields of psychiatry, oncology, and immunology. In a study by Sakamoto, K., et al. [37], the researchers combined AF2 with molecular dynamics (MD) simulation techniques to construct models of the VIPR2/KS-133 and VIPR2/vasoactive intestinal peptide (VIP) complex and to understand their binding modes. The VIPR2/KS-133 and VIPR2/VIP complex models were constructed using AF2 and molecular dynamic simulations.

### 5.2. Biomarker Discovery

Novel protein structure prediction algorithms provide information about the proteins’ structures that previously resisted attempts at experimental structure determination. A study published in Chemical Science applied AlphaFold to identify a new drug for hepatocellular carcinoma (HCC), the most common form of primary liver cancer [38]. This study used AlphaFold to predict the structure of CDK20 (Cyclin-Dependent Kinase 20), which is involved in cell cycle regulation; its abnormal activity can lead to uncontrolled cell growth, a hallmark of cancer. The researchers then identified potential inhibitor molecules using AI platforms developed by Insilico Medicine. They synthesized and tested these molecules, finding one, ISM042-2-048, with promising inhibitory activity against CDK20.

## 6. Comparative Analysis of Protein Structure Prediction Algorithms and Tools

### 6.1. Overview of the AlphaFold2 Algorithm

AlphaFold2 (AF2) is a state-of-the-art computational framework specifically designed to predict the three-dimensional structures of proteins. It uses a combination of sequence and structural databases to gather the necessary information for its predictions. Sequence databases such as UniRef90, BFD, and the Mgnify microbiome database [39] provide access to amino acid sequences used to build a multiple sequence alignment (MSA) for the query sequence; AF2 then uses the experimental structures from the PDB [24] to train the “structural module” that builds the final model. MSA of the sequences of the query homologs is used to predict pairwise distances between residues (a distance map), which are later refined in several rounds of iterations reconciling initial distance predictions with the constraints of the subsequent models of the query. AF2’s architecture and training methodology contributed to its high accuracy in 3D protein structure prediction and allowed it to dramatically improve the quality of protein structure predictions. At the same time, its singular focus on structure prediction and extensive use of multiple MSAs may have limited its ability to predict changes to structure caused by small changes in sequence (single point mutations) and affected its accuracy in predictions for “orphans”, proteins with few or no known homologs (Figure 3).

### 6.2. Overview of the ESMFold Algorithm

The ESMFold model [8] is built upon a BERT-like architecture, which is a type of large language model that utilizes stacked Transformer encoder layers. It is trained using a technique known as masked residue prediction, where certain amino acids in the protein sequence are hidden from the model during training, forcing it to predict these residues based on the surrounding context. This training process enables ESMFold to develop intricate internal representations of protein sequences. A notable feature of the ESM language model is its ability to infer structural information from protein sequences without relying on MSAs or known protein homologies.

The model’s attention maps, derived from sequence embeddings, are used to predict the contact map. This capability is based solely on the amino acid sequence of the protein, making ESMFold a valuable tool for studying proteins that are difficult to analyze using traditional methods that depend on evolutionary comparisons (Figure 4).

### 6.3. Overview of the RoseTTAFold Algorithm

Developed by David Baker’s group at the Institute for Protein Design at the University of Washington, RoseTTAFold [7] is an extension of the older Rosetta family of tools, enhanced by the deep learning technology. It employs a unique ‘three-track’ neural network and integrates three types of information: the sequential patterns in proteins, the interplay between amino acids, and the probable three-dimensional configurations. RoseTTAFold has recently been updated to model complete biological assemblies, including a range of biomolecules such as proteins, DNA, and RNA. This enhancement broadens the potential uses of protein structure prediction algorithms [40].

### 6.4. Overview of the OpenFold Algorithm

The OpenFold Consortium introduced OpenFold, an open-source, trainable version of AF2, alongside OpenProteinSet, a database of 5 million diverse MSAs. This eliminates the massive computational barrier—millions of CPU hours—required for large-scale training. When trained from scratch using OpenProteinSet, OpenFold matches AF2’s prediction quality but offers advantages like faster processing, lower memory usage for handling longer proteins on a single GPU, and compatibility with the widely used PyTorch machine learning framework. This makes OpenFold easily accessible to a broad developer community [9].

Using OpenFold, researchers explored the model’s protein-folding learning process, identifying distinct behavioral phases during intermediate training stages. They discovered that OpenFold learns spatial dimensions and structural elements in an interleaved fashion. With OpenFold achieving 90% accuracy in just 3% of the training time as AF2, its retraining on pruned data sets showcased robustness and varied generalization capabilities. Training on smaller, diverse data sets further enhanced OpenFold’s performance. These findings provide valuable insights into AF2-type models and pave the way for advancements in biomolecular modeling algorithms.

### 6.5. Comparing AlphaFold2 vs. ESMFold vs. RoseTTAFold vs. OpenFold

In protein structure prediction, utilizing individual sequences without relying on co-evolutionary data like MSA emerges as a promising strategy. This method potentially eliminates the time needed for homology searches and MSA building and may enhance prediction accuracy for orphan proteins. Although explored in earlier research by Chowdhury et al. and Wang et al. [41,42], the results were initially less than ideal. However, recent ESMFold results indicate that larger pre-trained models alongside techniques inspired by AF2’s distillation method can enhance prediction accuracies. This improvement is attributed to two primary factors. First, the size of the sequence pre-trained models has been significantly increased, with ESMFold now using a 15B model that encapsulates more co-evolutionary information. Second, instead of employing self-distillation, a technique known as AF2 distillation has been adopted. In this approach, AF2 is utilized to perform structure predictions on a large sequence database, and the predicted structures are then used as training data for ESMFold. This innovative method of utilizing AlphaFold2’s predictive power to enrich the training data has contributed to the enhanced performance of ESMFold in protein structure prediction. For instance, ESMFold, with fewer parameters, predicts a protein with 384 residues in just 14.2 s on a single NVIDIA V100 GPU, about 6 times faster than AF2.

The strategies employed by AF2, ESMFold, RoseTTAFold and OpenFold in protein structure prediction offer distinct advantages and limitations. ESMFold’s approach of using individual sequences for predictions is time-efficient and particularly beneficial for orphan proteins, which lack homologs in current databases. ESMFold, demonstrates a significant speed advantage over AF2, enabling the rapid construction of predicted structures, a crucial factor given the vast amount of available sequence data.

On the other hand, AF2’s methodology, as summarized in the overview, leverages MSA and structural databases to interpret coevolutionary correlations between mutations for its predictions. However, this approach may pose challenges in handling novel single-point mutations or orphan proteins and concerns regarding data leakage in evaluation data sets.

RoseTTAFold can predict protein–nucleic acid complexes, though its precision in this area is not as high as when dealing with protein structures alone. To enhance this capability, the RoseTTAFoldNA extension has been developed, specifically focusing on improving the predictions of protein-nucleic acid complexes [43].

The contrasting approaches among AF2, ESMFold, RoseTTAFold and OpenFold highlight the trade-offs between prediction speed and accuracy and need for additional input data (Table 2). We compared the algorithms used by AF2, ESM2 and OpenFold focusing on the input and frameworks in Appendix A.

### 6.6. AI Tools in Protein Sequence Generation and Structure Design

Generating protein sequences for novel proteins with designed structures and functions is an interesting extension of the protein structure prediction problem, pave the way for novel treatment options. Computational models like ProGen, ProteinMPNN, EvoDiff, and RFDiffusion are being leveraged to accelerate this process.

ProGen, short for Protein Generator, is a protein language model developed by Salesforce AI Research that generates protein sequences with predictable functions [10]. ProGen, a 12-billion-parameter neural network, generates protein sequences for specific biological functions. It uses functional tags from the Pfam database and is trained on 280 million protein sequences. Researchers have fine-tuned it with distinct enzyme families, resulting in millions of sequences that closely resemble natural enzymes.

ProteinMPNN [11] is a deep learning algorithm designed for protein sequence design. It extends the message-passing neural network (MPNN) framework, which is a machine learning technique that can predict the properties of properties by simulating how residues send and receive information to and from their neighbors. ProteinMPNN has been extended to design protein–nucleic acids and protein–small molecules, which will greatly increase its utility.

EvoDiff, a general-purpose diffusion framework developed by Microsoft [12], is tailored for generating protein sequences and evolutionary alignments. It capitalizes on large-scale evolutionary data and the unique conditioning strengths of diffusion models. A prominent attribute of EvoDiff is its ability to generate proteins based solely on their sequence information. This streamlines the protein design process, as it negates the necessity for comprehensive structural data.

RFDiffusion, used in protein engineering, leverages generative AI to generate novel protein sequences and structures. RFDiffusion is a generative model that creates protein sequences and structures using denoising diffusion probabilistic models [13]. RFDiffusion takes a unique approach by adding Gaussian noise independently to the rotation and translation matrices of the protein’s 3D coordinates to generate training data. This results in a model with higher-dimensional representation capabilities and global rotational invariance, which in turn enables more stable model training. During the denoising process, each step of the model predicts the structure after local denoising for the subsequent step. This predicted structure then serves as the initial coordinate and structural template for further predictions. Ablation studies have confirmed the importance of these templates in generating high-quality protein structures. Additionally, RFDiffusion incorporates a sequence information channel. Sequences that are masked during the diffusion process gradually recover, mirroring a training task approach from a previous model, RFjoint [46]. This allows RFDiffusion to predict amino acid distributions at masked string positions, and it has led to speculation that RFDiffusion might essentially be an evolved version of RFjoint, enhanced by adding structural template noise. Moreover, RFDiffusion offers different versions tailored to specific tasks, such as fixing known or functional segment structures, broadening its applicability in protein research (Figure 5).

### 6.7. AI Tools in Docking Used in Drug Discovery

Docking, a computational strategy, predicts how two molecules form a stable complex and is usually separated into ligand docking and protein–protein docking. AI boosts this process’s speed and precision. Deep Docking (DD), an AI enabled methodology for virtual screening of ultra-large chemical libraries, significantly accelerates structure-based virtual screening. DD iteratively docks subsets of a chemical library, synchronized with ligand-based predictions, to enhance virtual hit enrichment without substantial loss of potential drug candidates [47]. DiffDock, an AI-driven tool from MIT, frames molecular docking as a generative modeling problem. It maps the manifold of ligand poses to the product space of the degrees of freedom involved in docking (translational, rotational, and torsional) and develops an efficient diffusion process on this space [14].

## 7. Generative AI: A Catalyst in Cancer Drug Development

Generative AI has emerged as a transformative force in the life sciences sector (Figure 6), powering innovative research, optimizing workflows, and providing new insights. Its applications are extensive and varied: We previously discussed de novo design of proteins [48,49], the creation of novel antibodies [50], and the building of comprehensive models for single-cell multi-omics [51], which can provide a deeper understanding of the cellular heterogeneity in tumors and inform the development of personalized cancer treatments.

Moreover, generative AI also plays a role in genomic variant effect prediction [52] and identifying statistical patterns in DNA sequences [53], which can help in understanding the genetic basis of cancer. It is instrumental in predicting and reconstructing the evolution of viruses [54], thus offering valuable insights for epidemiology and vaccine development. Additionally, this technology can generate synthetic data to augment existing data sets [55], providing a valuable resource for researchers and scientists. Even in the realm of data visualization, generative AI can be used for text-to-image generation [56,57], translating complex textual descriptions into accurate, understandable biological images. Overall, by enabling a deeper understanding of biological systems and accelerating the discovery process, it holds great promise in advancing the fight against cancer.

## 8. Beyond Cancer: AI Driven Drug Discovery for Other Diseases

Researchers at MIT used AI to discover a new antibiotic, named “halicin”, effective against E. coli and drug-resistant Acinetobacter baumannii. They trained a neural network on a data set of 2335 molecules that inhibit *E. coli* and refined the model with feature engineering and other techniques. The AI model analyzed over 107 million molecules to identify potential antibiotics against E. coli. Using the model’s predictions, a shortlist of the most promising candidates was created and empirically tested for their antibacterial properties [58].

In early 2020, Exscientia reported the initiation of Phase 1 clinical trials for DSP-1181, a compound designed to address obsessive-compulsive disorder. Developed through AI, the compound was identified by screening chemical libraries for the most pertinent candidates. DSP-1181 is reported to be the first drug of its kind to reach the clinical trial stage [59]. It was the first AI-designed drug candidate to enter clinical trials, which was a pivotal moment in AI drug discovery.

In February 2023, Insilico Medicine received the FDA’s first-ever Orphan Drug Designation for a medication discovered and developed through AI technology. The drug, named INS018_055, is designed as a small molecule inhibitor for treating idiopathic pulmonary fibrosis (IPF). Utilizing their proprietary Pharma.AI platform, Insilico not only identified a new target but also generated innovative small molecules. Following the successful completion of Phase 0 and Phase I safety studies, the drug has now advanced to multi-regional Phase II clinical trials in the United States and China [60,61].

AI has been employed in the quest for COVID-19 therapeutics. Researchers combined AI with fragment-based drug design to speed up the identification of potential drugs against SARS-CoV-2. Using a molecular library of known SARS-3CLpro inhibitors; they utilized AI to generate new compounds targeting the virus’s essential 3CL protease. These AI-generated molecules were then screened for their ability to inhibit viral replication by binding covalently to the 3CL protease [46]. In a different study, deep neural networks were used to create new small molecules targeting SARS-CoV-2’s 3CL protease. Utilizing transfer and reinforcement learning, the generative model was fine-tuned to focus on known protease inhibitors. The training data came from the ChEMBL database of viral protease inhibitors [62].

## 9. Limitations and Challenges

AF2 demonstrates high accuracy in predicting the three-dimensional structures of proteins, particularly when sequences of multiple homologs are available in sequence databases to construct an MSA. However, along with other AI tools, it requires substantial computational resources. This can limit its accessibility for researchers with limited computational capabilities [63]. The integration of AI into the drug discovery process also presents regulatory and implementation challenges. These include ensuring data privacy and security, validating the effectiveness of AI-based predictions, and adapting existing workflows and systems to incorporate AI tools [64].

The challenge of ligand-induced folding, especially in regions of proteins that are intrinsically disordered or “floppy”, has been a significant obstacle in drug discovery and development [35]. AF2, despite its groundbreaking contributions to protein structure prediction, primarily predicts a single static state of a protein. This approach overlooks the dynamic conformational changes critical for enzyme function and drug interaction, such as adjustments at the binding site or domain shifts. These dynamic changes are essential for understanding how a protein functions in its natural environment and how it interacts with potential drug molecules [35]. The study by Fernández [65] introduced a deep learning approach to address the challenge of binding-induced folding in a protein’s intrinsically disordered regions. However, the conformational change induced by drug folding in proteins with multiple domains or involved in protein–protein interactions remains a challenge [66].

Finally, it is important to note that in drug discovery, understanding a protein’s structure, while insightful, is seldom the primary bottleneck; the process is driven more by empirical data from assays, pharmacokinetics, metabolism, and toxicology, emphasizing that the success of drug discovery hinges on multiple factors.

As the field of protein structure prediction continues to evolve, it is expected that further advancements will be made in the accuracy and applicability of AI tools. These improvements may include enhanced performance on challenging protein classes and complexes, better handling of protein dynamics and conformational changes, and reduced reliance on homologous templates for accurate predictions [35]. Additionally, incorporating experimental constraints and other sources of information into the modeling process may help to increase the accuracy and reliability of structure predictions for a wider range of protein targets.

## 10. Conclusions

AI is poised to significantly transform the landscape of drug development, offering ways to streamline the process, reduce costs, and enhance success rates at various stages. The process started with AF2, which has achieved remarkable success in predicting protein structures, marking a milestone for AI applications in structural biology. By providing accurate predictions of protein structures, AF2 can accelerate the development of new cancer drugs and therapies, and more effectively identify and validate novel drug targets, particularly for those lacking substantial structural information. Perhaps more importantly, AF2 has inspired a wave of AI-driven tools in protein structure prediction, engineering, docking, and generating novel proteins with desired structures and functions. These tools exemplify the role of AI in advancing drug development by enabling the generation of novel protein sequences and structures, predicting the effects of genomic variants, and providing new insights into the mechanisms of cancer.

## Figures and Tables

**Figure 1 biomolecules-14-00339-f001:**
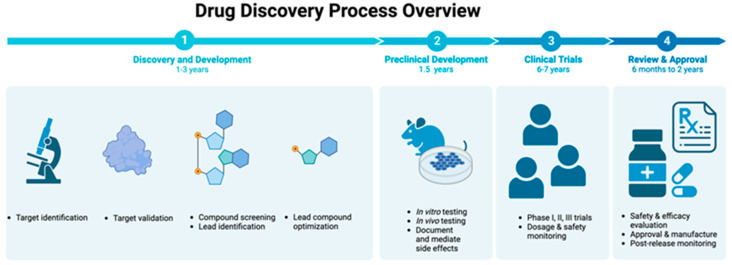
Stages of Drug Discovery Process: The drug discovery process comprises several critical stages. It begins with the “Discovery and Development” phase, where the focus is on target identification and validation. This stage involves screening potential compounds and further refining promising candidates through hit-to-lead development and lead optimization. Following this, the process moves to “Preclinical Development”, which includes a range of lab tests such as in vitro studies, animal model testing, and ADMET (Absorption, Distribution, Metabolism, Excretion, Toxicity) studies. Based on these results, a decision is made on whether to proceed to the next phase. “Clinical Trials” ensue, which are categorized into four phases: Phase I assesses safety and dosage; Phase II examines efficacy and side effects; Phase III involves larger studies to confirm efficacy and monitor adverse reactions; and the final stage is “Review and Approval”, which consists of a comprehensive regulatory review, culminating in market authorization and followed by post-marketing monitoring to ensure long-term safety and effectiveness, constituting a newly defined Phase IV.

**Figure 2 biomolecules-14-00339-f002:**
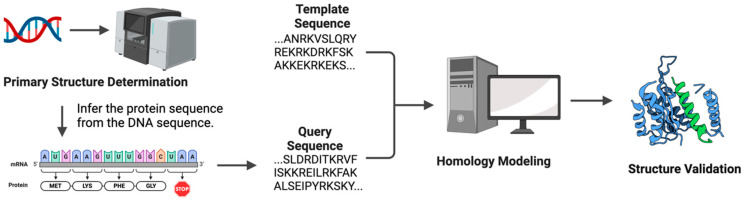
Stages of Protein Structure Prediction: The foundational stage involves determining the DNA sequence that encodes the protein of interest. The next step is to infer the protein sequence from the DNA sequence. Homology modeling uses known protein structures as templates to predict the structure of a protein with an unknown structure but similar sequence. Lastly, validation of structure ensures the predicted structure’s biological plausibility. This involves checks on stereochemical quality, energy evaluation, and comparison to known structural data.

**Figure 3 biomolecules-14-00339-f003:**
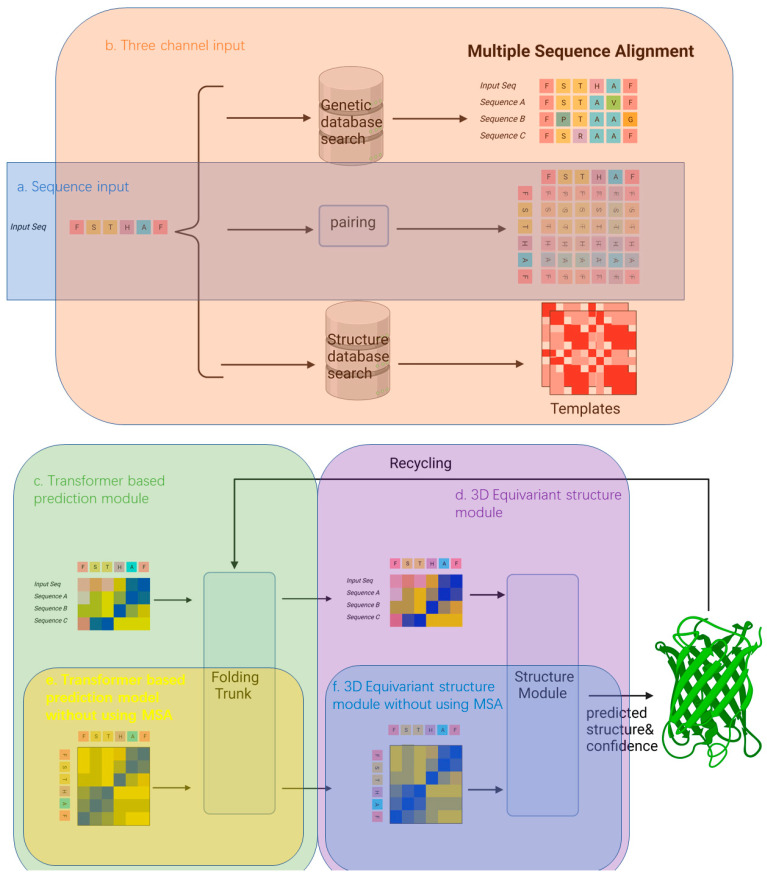
Model Architecture of AlphaFold2. The architecture of the AlphaFold2 model can be broadly divided into three parts: (1) Model Input (2) Evoformer (3) Structure module.

**Figure 4 biomolecules-14-00339-f004:**
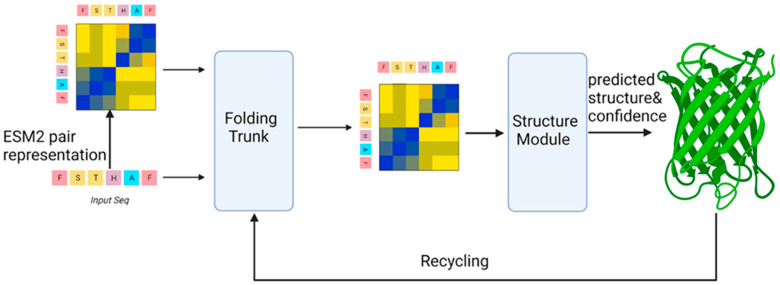
Model Architecture of ESMFold. The ESMFold model can be divided into four parts: data parsing, encoder (Folding Trunk), decoder (Structure Module), and the recycling phase.

**Figure 5 biomolecules-14-00339-f005:**
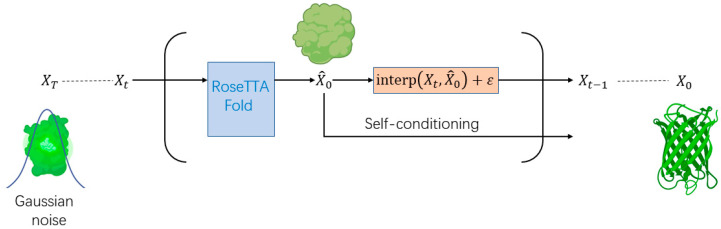
Model structure of RFDiffusion. Use of the diffusion model approach for training and fine-tuning the protein structure prediction model, enabling a more refined depiction of the hidden relationship between protein sequences and structures.

**Figure 6 biomolecules-14-00339-f006:**
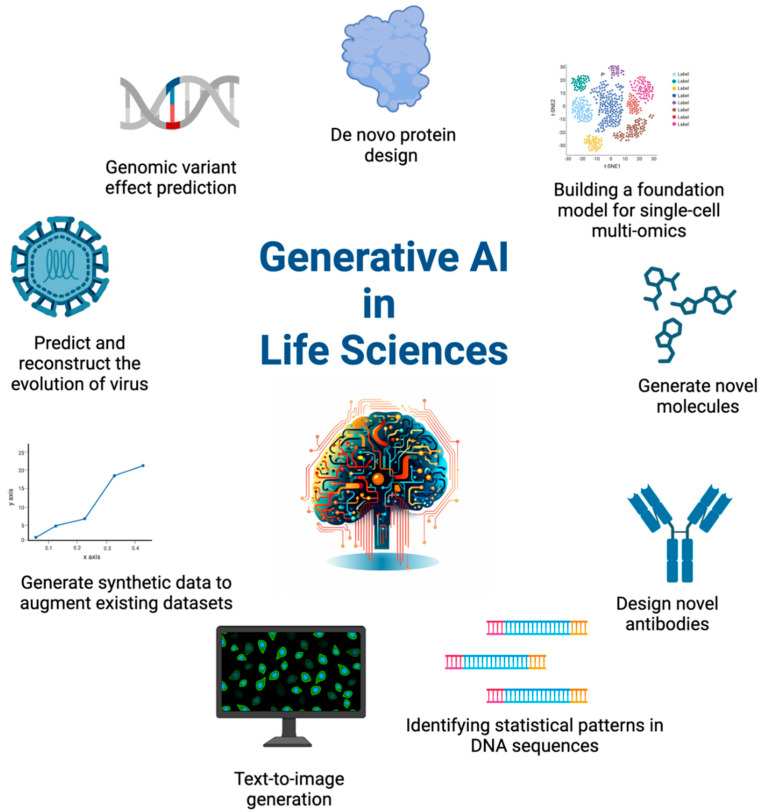
Generative AI in Life Sciences: A Comprehensive Overview of Applications and Innovations. Generative AI is revolutionizing various aspects of life sciences. It is accelerating drug discovery, aiding in antibody development, and enhancing single-cell multi-omics models for disease understanding. The technology also plays a role in personalized medicine, population genetics, and viral evolution. Beyond biology, it is pivotal in data science for generating synthetic data and in scientific visualization through text-to-image technologies. Overall, generative AI’s impact is expansive and transformative across life sciences.

**Table 1 biomolecules-14-00339-t001:** Publicly available protein structure data sets and their applications in different phases of drug discovery.

Resource	Utility	Data Repository and Reference
Protein Data Bank (PDB)	Provides 3D structures of proteins, nucleic acids, and complex assemblies which can be used for drug target identification, ligand design, and understanding protein–ligand interactions.	RCSB PDB [27]
AlphaFold Protein Structure Database	Contains protein structure predictions for entire proteomes of several organisms. It can be used for target identification and understanding protein function.	AlphaFold DB [25,26]
CASP (Critical Assessment of protein Structure Prediction)	Hosts protein structure prediction models from the CASP competition, useful for evaluating and improving structure prediction methods.	CASP [28]
SWISS-MODEL Repository	A database of annotated 3D protein structure models generated by the SWISS-MODEL homology-modeling pipeline, useful for structure prediction and drug design.	SWISS-MODEL [29]
ESM Metagenomic Atlas	The ESM Metagenomic Atlas displays more than 700 million predicted protein structures from microorganisms in environments like soil, seawater, and the human gut, accessible through an interactive page.	ESM Atlas [8]

**Table 2 biomolecules-14-00339-t002:** Capabilities of and differences between these four protein structure prediction models.

Model	Speed	Accuracy [44]	Use of MSA	Strength
AlphaFold2	Requires high-powered and high-capacity computing resources	AlphaFold2 attains a mean GDT-TS score of 73.06.	Yes, leverages MSA for rich evolutionary context	High accuracy
ESMFold	6× faster than a single AlphaFold2 model.	ESMFold attains a mean GDT-TS score of 61.62.	No, predicts structures from a single sequence	Fast prediction speed
RoseTTAFold	Vary depending on the specific protein and computational resources, compared to AlphaFold2.	In over 80% of cases, RoseTTAFold’s performance was lower than ESMFold, with the latter achieving a higher mean GDT-TS score.	Yes, uses MSAs	predicting protein complexes with RNA or DNA
OpenFold	Slightly faster than AlphaFold2 [45].		Yes, uses MSAs	Allows for application-specific training

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
