# Peer review of "Advances in AI for Protein Structure Prediction: Implications for Cancer Drug Discovery and Development"

_biomolecules, 2024, doi:10.3390/biom14030339_

Round 1
Reviewer 1 Report
Comments and Suggestions for Authors
By now there are many reviews on AI approaches to structure prediction and AlphaFold2 in particular, assessing their impact in drug discovery and lead optimization. One of the better ones is
Neera Borkakoti and Janet M. Thornton: “Alphafold2 protein structure prediction: Implications for drug discovery”. Curr Opin Struct Biol. 78, 102526 (2023)
Unless the authors decide to address the limitations of such approaches, I don't really see the need for yet another review. The authors should deal with the AI approaches to drug-induced structural changes in the target protein (structural adaptation). Binding-induced folding, particularly in floppy regions of the target protein seems to be the Achilles' heel of the AI approaches currently adopted. A good starting point would be
Ariel Fernandez: “Perspective: Artificial intelligence teaches drugs to target proteins by tackling the induced folding problem”. Molecular Pharmaceutics (ACS) 17, 2761-2767 (2020)
and I assume there are other recent contributions along these lines. The drug-induced folding problem seems more relevant to drug design and development than the folding problem itself, and signals crucial limitations on AI approaches when adopted by the pharmaceutical industry. Unless the authors address these aspects, I don't see why this review may be needed.
Author Response
Response to Reviewer 1 Comments
- Summary
Dear reviewer, we thank you very much for your feedback on our manuscript. We appreciate your suggestions for the additional references, such as Neera Borkakoti and Janet M. Thornton or Ariel Fernandez papers. We have not included them in our review.
Once again, we thank you for your constructive comments and for guiding us toward enhancing the quality and relevance of our review.
- Point-by-point response to Comments and Suggestions for Authors
Reviewer Comment: By now there are many reviews on AI approaches to structure prediction and AlphaFold2 in particular, assessing their impact in drug discovery and lead optimization.
Response: We agree, which is why we focused on a) new developments, which have not yet been comprehensively synthesized in the literature b) limitations of individual methods that result from their specific architectures.
Reviewer Comment: Neera Borkakoti and Janet M. Thornton’s review is one of the better ones in addressing these topics.
Response: We have added Borkakoti and Thornton’s article in our review as reference #34, specifically in line 161-164, 420-427, 436-439. In the revisions, we specifically discuss AI methods in the accurate modeling of drug-induced structural changes.
Reviewer Comment: The authors should address the limitations of AI approaches to drug-induced structural changes in target proteins (structural adaptation).
Response: We appreciate this suggestion. In our revised manuscript, we added a section ‘AI Methodologies and Limitations in Modeling Drug-Induced Structural Changes’ under section ‘Limitations and Challenges’ in line 420-430 highlights in blue, focused on discussing the limitations of AI in modeling drug-induced structural changes.
Reviewer Comment: Binding-induced folding, particularly in floppy regions of the target protein seems to be the Achilles' heel of the Al approaches currently adopted. A good starting point would be
Ariel Fernandez: "Perspective: Artificial intelligence teaches drugs to target proteins by tackling the induced folding problem". Molecular Pharmaceutics (ACS) 17, 2761-2767 (2020)
and I assume there are other recent contributions along these lines. The drug-induced folding problem seems more relevant to drug design and development than the folding problem itself and signals crucial limitations on Al approaches when adopted by the pharmaceutical industry.
Response: We agree with the reviewer’s comment. We included a critical analysis of how current AI methods in section Advancements in AI for Overcoming Drug- ‘Induced Folding in Protein Design’ under section ‘Target Identification’. We aim to highlight the necessity for improved modeling of these ‘floppy’ regions and discuss potential solutions that are being explored in the field.
We added the reference to the Fernandez’s manuscript as referenced #64. Our revised manuscript emphasized the aspect of drug induced structural changes and elaborated it in line 427-428 and highlight in blue.
Reviewer Comment: The review may not be needed unless it addresses these crucial limitations on AI approaches.
Response: We enhanced our review to address the critical limitations of AI in the context of drug design, ensuring that our work adds value to the scientific community and to those in the pharmaceutical industry who might employ these AI techniques.
Conclusion: We are grateful for your constructive criticism and guidance. We believe that our review is now better and fills a gap in the current literature.
Reviewer 2 Report
Comments and Suggestions for Authors
The manuscript reviewed the various AI developments, from target ID to drug design for the treatment of cancer. Initially, the authors introduce the traditional stages for structure prediction, starting from sequence analysis and finally, structure validation. The results of these preliminary studies are a series of databases that have helped drive the mechanistic studies of disease progression. The authors introduced many AI tools available and discussed their application in their respective fields. Further, the authors focus on the AI-based models alphafold2, ESMFold, RoseTTAFold, and Openfold and go into the pitfalls of each technique and their role in target identification and drug design.
The authors presented the information in a clear and easy-to-follow format. They also compared the various models and highlighted some pitfalls. Furthermore, the authors provide a complete view of technology's current limitations and provide cases where the technology has been successfully utilized. I feel the authors could have mentioned docking tools used in drug discovery, like Schrodinger’s Maestro software or Pymol.
Overall, the review presents a broad overview of what is currently out there and provides a good review of each model. As this technology matures and more researchers turn to in silico applications, there will be a need to identify what model best suits a specific question.
Comments on the Quality of English LanguageOverall I think that the paper should be accepted in its present form with little final edits to the English language.
Author Response
Response to Reviewer 2 Comments
- Summary
We thank you very much for your feedback on our manuscript. We are pleased to hear that you found the presentation of AI developments in cancer treatment, from target identification to drug design, to be clear and informative.
Following your suggestions, we include a discussion on AI docking tools used in drug discovery.
Your comment on the quality of the English language and the readiness of the manuscript for publication with minor edits is greatly appreciated. We carefully reviewed the manuscript to identify and correct any remaining language issues to ensure clarity and readability.
- Point-by-point response to Comments and Suggestions for Authors
Reviewer Comment: I feel the authors could have mentioned docking tools used in drug discovery, like Schrodinger's Maestro software or Pymol.
Response: Thank you for the recommendation to address the inclusion of docking tools. We recognize the importance of this aspect and have added a dedicated section to our manuscript to offer a comprehensive view of the computational methods used in drug discovery.
Titled 'AI Tools in Docking Used in Drug Discovery' (line 346 – 356), this section falls within the broader 'Comparative Analysis of Protein Structure Prediction Algorithms and Tools'. Deep Docking (DD) is an AI-based approach that allows for the virtual screening of exceptionally large chemical libraries with increased speed. It works by iteratively docking subsets of a chemical library while integrating ligand-based predictions to improve the identification of potential hits, ensuring minimal loss of viable drug candidates. Another innovative AI tool, DiffDock, developed by MIT, approaches molecular docking through generative modeling. It treats the docking process as a mapping problem involving the ligand's translational, rotational, and torsional degrees of freedom, employing a diffusion process to efficiently explore this space. These advancements demonstrate AI's transformative potential in streamlining the drug discovery process by enhancing virtual screening and docking methodologies.
Reviewer Comment: Overall I think that the paper should be accepted in its present form with little final edits to the English language.
Response: We thank for your positive assessment of our paper. In our author list includes native English speakers who had meticulously reviewed and refined the manuscript to correct any remaining linguistic errors. For this purpose, all the revised part are highlight in orange.
We had addressed each of the points you have raised in our revision and look forward to the improved manuscript contributing meaningfully to the field.

Round 2
Reviewer 1 Report
Comments and Suggestions for Authors
I have read the revision and I am satisfied with the changes. The review incorporates some novelty that makes it worth publishing, distinguishing itself from other reviews on the topic.